# INViTE: INTERPRET AND CONTROL VISION-LANGUAGE MODELS WITH TEXT EXPLANATIONS

**Haozhe Chen[1], Junfeng Yang[1], Carl Vondrick[1], Chengzhi Mao[123]**
Columbia University[1], Mila[2], McGill University[3]
`hc3295@columbia.edu`, `{junfeng,vondrick,mcz}@cs.columbia.edu`

## ABSTRACT

Large-scale pre-trained vision foundation models, such as CLIP, have become de facto backbones for various vision tasks. However, due to their black-box nature, understanding the underlying rules behind these models' predictions and controlling model behaviors have remained open challenges. We present INViTE: a framework for INterpreting Vision Transformer's latent tokens with Text Explanations. Given a latent token, INViTE retains its semantic information to the final layer using transformer's local operations and retrieves the closest text for explanation. IN-ViTE enables understanding of model visual reasoning procedure without needing additional model training or data collection. Based on the obtained interpretations, INViTE allows for model editing that controls model reasoning behaviors and improves model robustness against biases and spurious correlations. Our code is available at `https://github.com/tonychenxyz/vit-interpret`.

## 1 INTRODUCTION

With the advent of large-scale foundation models, transformers have become the backbone for machine learning tasks (Zeng et al., 2022; Rombach et al., 2022; Girdhar et al., 2023; Devlin et al., 2018; Kirillov et al., 2023). Recent advancements in large-scale vision-language transformers, such as CLIP (Radford et al., 2021; Jia et al., 2021) and FLAVA (Singh et al., 2022), have enhanced robustness and generalizability. These models leverage visual tokens to represent images and enable visual reasoning through attention operation on different tokens (Dosovitskiy et al., 2020). However, since those latent tokens are continuous, high-dimensional vectors (Geirhos et al., 2020; Mao et al.), it is challenging to interpret and understand the reasoning procedure of transformers.

Interpretable models offer tremendous benefits, fostering better understanding and trust among users (Hendricks et al., 2016), particularly in critical applications like medical diagnosis and treatment (Pereira et al., 2018). Moreover, interpretable models enable users to control and intervene, improving human-AI interaction.

Previous attempts at interpreting transformer models have primarily focused on generating saliency maps (Ancona et al., 2018; Selvaraju et al., 2016), lacking concept-level explanation for the latent embeddings. Although efforts such as NetDissect (Bau et al., 2017) connect concepts and neuron activations using labeled attribute datasets, these approaches require manual annotation, hindering generalization in open-world settings. Other work generates language rationales for the model's decision, yet the prediction might not be made based on the provided rationales (Mao et al., 2022b; Hendricks et al., 2016; Kim et al., 2018b).

In this paper, we propose INViTE: an approach to INterpret Vision Transformer *latent* tokens using Text Explanations. In the established CLIP model, a text description is retrieved from a set of provided vocabulary for entire image based on similarity between image and text representation. We extend this procedure to retrieve text descriptions for all latent tokens. Figure 1 shows example text explanations and attention heat-maps associated with the interpreted tokens in an ImageNet image. The interpretations are consistent with image areas that the tokens attend to and reveal CLIP's reasoning process of assembling parts into a whole concept.

Retrieving natural language descriptions for latent tokens is challenging since transformers' latent space does not directly associate with word embeddings. Our key hypothesis is that latent tokens in

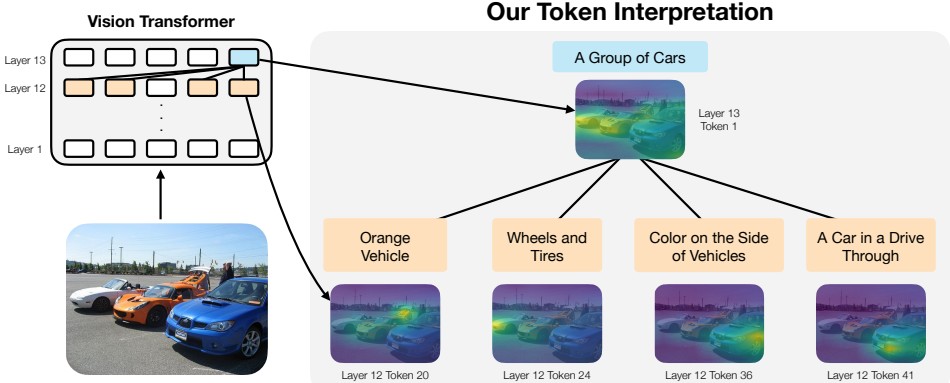

Figure 1: Interpreting visual reasoning in transformer. Our approach allows text interpretation for latent tokens without any training or data collection. The diagram shows sample interpretations and image areas the tokens attend to. Our interpretations also reveal transformer's hierarchical reasoning process between two layers.

transformers maintain same semantic information in the subsequent layers *when they do not attend to other tokens*. Our approach capitalizes on this property to interpret latent tokens by mapping latent embeddings to the final layer through forward propagation without self-attention operations.

A pivotal advantage of our framework is its ability to leverage an open-world vocabulary to explain latent token embeddings. By returning a language description for each latent token, our method directly elucidates the concepts learned within the transformers. Importantly, our approach does not require data collection, model retraining, or modification of architectures (Kim et al., 2018a; Koh et al., 2020).

Visualization and empirical experiments show that our method produces text explanations for latent tokens more aligned with the ground truth. Based on the provided interpretations, our framework enables users to intervene and control the model, such as fixing typographical attacks, removing spurious correlations, and replacing entities. Our approach works on open-world datasets, such as domain-specific satellite data, without additional training and data collection.

## 2 RELATED WORK

**Transformer.** The transformer architecture was first proposed in natural language processing by Vaswani et al. (2017). Due to its flexibility and versatility, it has become the standard architecture in various domains (Devlin et al., 2018; Girdhar et al., 2023; Dosovitskiy et al., 2020). The vision transformer (ViT) (Dosovitskiy et al., 2020) was the first work to adapt the transformer for vision tasks. Consequently, there is a growing need to develop tools to understand and interpret transformer-based architectures. The success of transformers relies on the attention operation (Bahdanau et al., 2014), which computes each token in the model by attending to a few other tokens. Recently, transformer becomes the backbone for vision foundation models (Zhu et al., 2023; Radford et al., 2021; Li et al., 2023; Zhang et al., 2022); however, the internal mechanism of transformer on visual reasoning still remains unknown.

**Model Interpretation.** Works such as Netdisssect in (Bau et al., 2017) interpret neural networks with top-activating image regions of individual neurons. Nguyen et al. (2017; 2016); Olah et al. (2017) find the inputs that maximize the outputs of features. Other works (Mahendran & Vedaldi, 2015; Zeiler & Fergus, 2014) invert neural feature maps to the pixel space for visualization. Gradient-based feature visualizations (Lundberg & Lee, 2017; Ribeiro et al., 2016; Simonyan et al., 2014; Shrikumar et al., 2017; Zeiler & Fergus, 2014; Smilkov et al., 2017; Selvaraju et al., 2016) find the input regions in images that are crucial to making decisions. Zou et al. (2023) establishes an AI transparency system that places representation at the center of analysis. Chen et al. (2023) builds a sparse representation space to enhance explainbility of model decisions.

Recent works seek to interpret models with natural language. MILAN (Hernandez et al., 2022) finds natural language descriptions by maximizing the pointwise mutual information between input regions and human annotations. However, it requires additional data collection and training, thus cannot

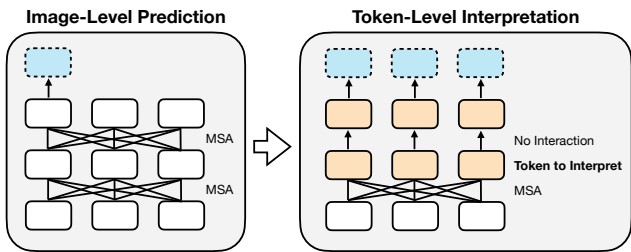

Figure 2: Latent token interpretation through disabling self-attention. Latent tokens in transformers will retain the semantic information in the subsequent layers when they do not attend to other tokens.

extend easily to open-world and domain specific tasks. Kim et al. (2018a) obtains concept activation vectors through learning on curated datasets. Other approaches (Mao et al., 2022b; Hendricks et al., 2016; Hernandez et al., 2022) generate text explanations for model predictions, but these methods do not guarantee to represent actual reasons behind model predictions. Koh et al. (2020) interprets model prediction to a predefined set of concepts, which prevent its generalization to the open world. These methods also often develops on CNN and fail to provide token-level understanding of transformers. Attention saliency maps exploits transformers' attention-based architectures by associating latent tokens with image areas they pay attention to (Caron et al., 2021; Abnar & Zuidema, 2020). However, attention visualizations do not provide concept level interpretations that explain models' higher level reasoning processes such as abstractions. Concurrent to our work, Gandelsman et al. (2023) provides conceptual interpretations for transformers through decomposing contribution factors to final outputs.

**Model Control.** Deep models are often repurposed through techniques like transfer learning (Kornblith et al., 2019; Ying et al., 2018) or fine-tuning. In the context of vision foundation models, visual prompting has been used to adapt models (Jia et al., 2022; Sandler et al., 2022). However, these approaches typically require access to a carefully tailored dataset specific to the task. For instance, to reduce bias, the dataset needs to be augmented with various text types to encourage the model to disregard them (Shetty et al., 2019; Mao et al., 2021; Bissoto et al., 2020). However, engineering such training data can be challenging and may not achieve perfection (Ponce et al., 2006; Torralba & Efros, 2011). Previous works studied controlling and rewriting generative models (Bau et al., 2020) and discriminative models (Santurkar et al., 2021) through modifying the *weights*. In contrast, our interpretation framework enables the model to directly modify the *representations* without altering the model weights during inference.

## 3 METHOD

### 3.1 PRELIMINARIES: ZERO-SHOT LANGUAGE RETRIEVAL

One widely used type of pretrained vision model includes the large-scale image-language pretrained models, such as CLIP (Radford et al., 2021). These models align visual and language representations through contrastive loss during training. They have demonstrated the ability to recognize objects in an open-world setting and exhibit superior robustness and generalization. Consequently, they serve as the backbone for various tasks, including diffusion (Rombach et al., 2022), robust visual recognition (Mao et al., 2022a), video perception (Xu et al., 2021), and object detection (Gu et al., 2021).

Vision-language models are often trained on pairs of images and corresponding texts, denoted as $(\mathbf{x}_i, \mathbf{t}_i)$. The image encoder is represented as $F_\theta$, which uses the final CLS token as the output. The text encoder is denoted as $T$. The model is trained using contrastive loss to align the image feature $F_\theta(\mathbf{x}_i)$ with the corresponding text feature $T(\mathbf{t}_i)$. After training, the model can perform visual recognition through nearest neighbor retrieval in the embedding space. From all possible descriptions in a set of provided vocabulary, it retrieves the nearest language descriptions as predictions for the visual input.

### 3.2 INTERPRETING LATENT TOKENS THROUGH DISABLING SELF-ATTENTION

We aim to gain insight into the latent tokens in a transformer model using natural language interpretation. A straightforward approach would be to train the latent tokens alongside their corresponding natural language explanations. This would allow us to create a space where we can retrieve from a set of provided vocabulary a nearest-neighbor text description to each latent token embedding. However,

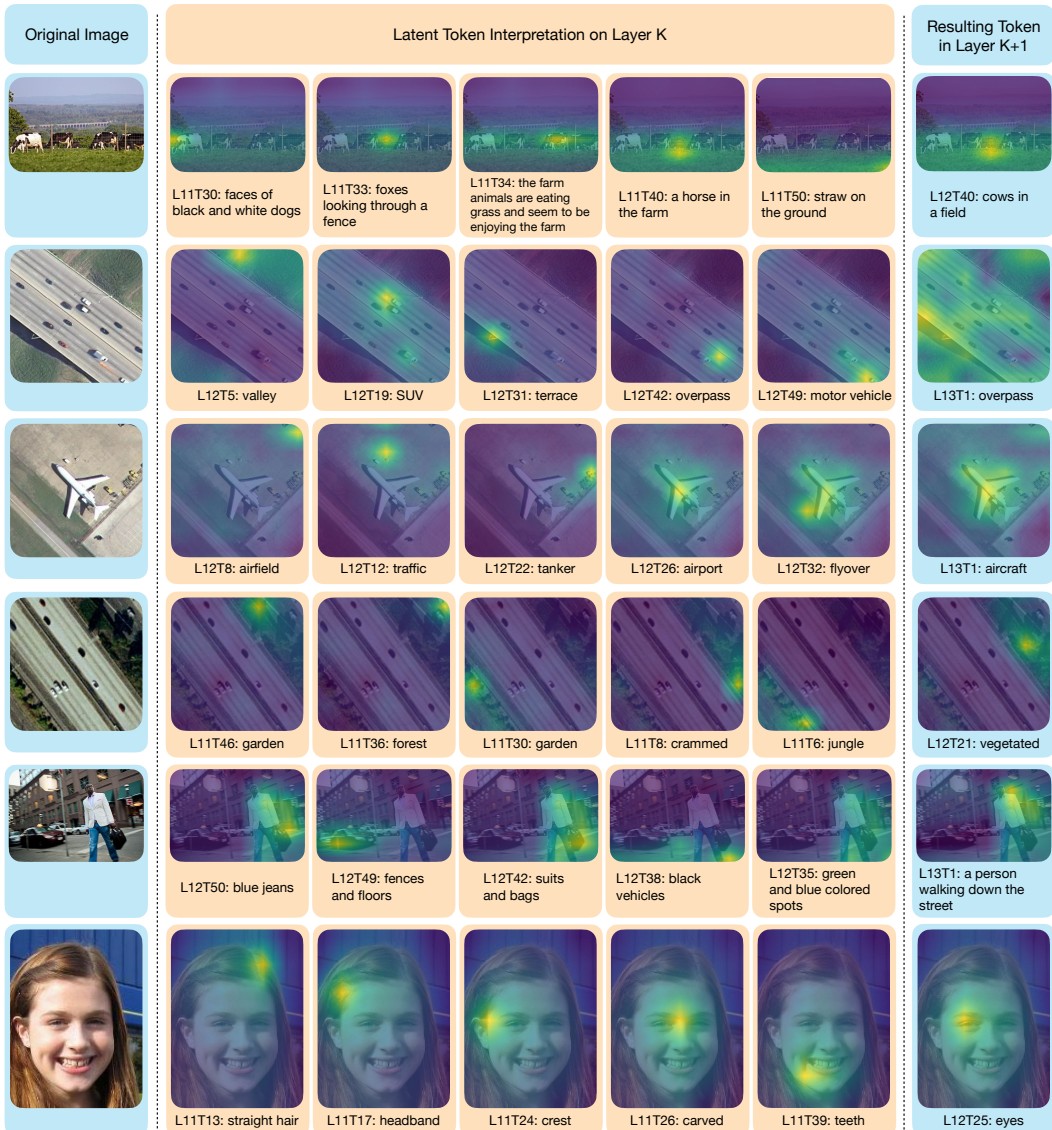

Figure 3: Examples of text interpretations obtained with INViTE. We show text explanation for $j$-th token in $i$-th layer as L$i$T$j$. To qualitatively verify correctness of these explanations, we visualize image areas most crucial to the token interpreted via rollout attention generated saliency map. Interpretations produced by our method are consistent with image areas most crucial to the tokens. Additional example explanations of tokens from earlier layers are provided in Appendix D.

unlike the text labels that are naturally paired with the entire image, we lack supervised text targets explicitly paired with latent tokens in each layer of transformers.

Ideally, if we could establish a mapping that converts the latent space to the final CLS output space, we could first find the final CLS output given the latent token query, and then use the final CLS output to retrieve language interpretations. However, since we lack data pairs of latent tokens and their representations in final CLS output space, we cannot directly train a model for such mapping.

We propose a method to map the latent tokens to the final CLS output space without any training or data collection, where we focus on repurposing the given transformer architecture. The Transformer architecture contains a sequence of blocks, where each block is built from a layer of multi-headed self-attention (MSA) and a multi-layer perceptron (MLP) with LayerNorm (LN). Residual connections are used inside the block.

The architecture of typical transformer can be formulated as this:

$$\mathbf{h}_0 = [\mathbf{x}_{cls}; \mathbf{x}^1; \mathbf{x}^2; \cdot; \mathbf{x}^L] + \mathbf{E}_{pos} \tag{1}$$

$$\mathbf{h}'_k = \text{MSA}(\text{LN}(\mathbf{h}_{k-1})) + \mathbf{h}_{k-1}, k = 1, ..., L \tag{2}$$

$$\mathbf{h}_k = \text{MLP}(\text{LN}(\mathbf{h}'_k)) + \mathbf{h}'_k, k = 1, ..., L \tag{3}$$

where we denote the CLS token as $\mathbf{x}_{cls}$, the $j$-th visual token input as $x_j$, the positional embedding as $\mathbf{E}_{pos}$, and the total number of block as $L$.

We divide the operations in the transformer into two types: in-token operations and inter-token operations. The mapping achieved by in-token operation can be denoted as a unary function $Z = f_u(\mathbf{x}_1)$, such as MLP, BN, and activation functions. The resulting feature $Z$ cannot contain additional information than that already contained in the input $\mathbf{x}_1$; thus, $Z$ does not have additional or different information than $\mathbf{x}_1$. Inter-token operations, specifically multi-headed self-attention (MSA), combine information from multiple tokens, which can be denoted as a multi-function $Z = f_m(\mathbf{x}_1, \mathbf{x}_2, ..., \mathbf{x}_n)$. The output token $Z$ thus can contain different information from each of $\mathbf{x}_1, \mathbf{x}_2, ..., \mathbf{x}_n$.

Motivated by this observation, we propose an approach to interpret the $j$-th token in the $i$-th layer. Our key insight is that, in the transformer computation, if a token does not interact with other tokens through self-attention, it will maintain the information local to the token in the subsequent layers.

We thus disable the attention operation from the $i$-th layer and above, retaining only the in-token operations. In MSA, this involves disabling the computation of keys (K) and queries (Q), and keeping only the token values (V) calculated from a linear transformation. We obtain the $j$-th output token in the final layer by forwarding the $j$-th token through the in-token operations. We can then use this output token's embedding to retrieve, from a set of provided vocabulary's embeddings, the corresponding language interpretation for the $j$-th token on the $i$-th layer. We will modify the computation of the features as follows:

$$\mathbf{h}'_k = f_V(\text{LN}(\mathbf{h}_{k-1})) + \mathbf{h}_{k-1}, k = i, ..., L \tag{4}$$

$$\mathbf{h}_k = \text{MLP}(\text{LN}(\mathbf{h}'_k)) + \mathbf{h}'_k, k = i, ..., L \tag{5}$$

The linear transformation operation for V in the MSA attention is denoted as $f_V$, and $i$ is the layer number we aim to interpret.

### 3.3 CONTROLLING MODEL BEHAVIOR THROUGH TOKEN REPLACEMENT

While previous works such as (Santurkar et al., 2021; Bau et al., 2020) control deep models by modifying their weights, we propose a method to control models by modifying their latent tokens. Guided by the text explanation of latent tokens, we can edit the model's reasoning procedure at inference time by replacing specific tokens with desired ones. Effective model editing based on INViTE text interpretations will also indicate the high quality of those interpretations. To demonstrate the effectiveness of our approach, we consider three tasks:

**Fixing Typographical Attacks.** Typographical attacks involve overlaying words on images to cause model mis-predictions, which has been successful against the CLIP model. Guided by INViTE interpretations, our approach first identifies latent tokens that interpret to descriptions such as *This is a text* and *This is a word*. We then replace these tokens with zero vectors. Since the dot product of the zero vector with all other tokens is zero, information from this token is not combined into other tokens in the next layer. The zero token will become a equal combination of all other tokens on the next layer since attention value goes through softmax. Our method intervenes in the transformer's reasoning process and removes the text's influence during inference requiring no additional data, training, or assumption about the content of the overlaid words.

**Intervening in the Reasoning Procedure.** We investigate whether INViTE interpretation explains the causal influence between a token and final image representation through examining whether our interpretation enables us to intervene in the transformer's reasoning process. For instance, when looking at a satellite image of cars on concrete roads, the existing CLIP model can reason that it is a highway. Our method first identifies the latent tokens that interpret to texts such as *This is a car* and *This is an automobile* in a highway image. Then, we extract tokens that interpret to texts such as *This is a plane* and *This is an aeroplane* in an airport image. Finally, we replace each car token with an airplane token, thereby intervening in the reasoning procedure of the transformer. A visualization of

Table 1: Fixing Typographical Attacks. The original CLIP model performs well on clean forest images, yet it is misled by typographical attacks of overlaying text *ocean* on images. Using INViTE interpretations, we can explicitly remove the latent tokens that represent text, which repairs attack in 97% of samples.

| | No Intervention | Random Intervention | Our Intervention | Our Intervention (RS) |
|---|---|---|---|---|
| % original image predicting forest ↑ | 94.00 | 97.00 | 97.00 | 97.00 |
| % original image predicting ocean ↓ | 5.00 | 3.00 | 0.00 | 0.00 |
| % attack image predicting forest ↑ | 1.00 | 17.00 | **98.00** | **98.00** |
| % attack image predicting ocean ↓ | 99.00 | 82.00 | **0.00** | **0.00** |

Table 2: Fixing Typographical Attacks. We followed experiment setup in Hernandez et al. (2022). With random smoothing, our method improves accuracy on attack images by 35.20%. In contrast, best improvement in Hernandez et al. (2022) is 4.9%. Editing based on INViTE interpretations also outperforms editing based on various saliency map methods.

| | Accuracy (original image) ↑ | Accuracy (attack image) ↑ |
|---|---|---|
| No Intervention ( (Hernandez et al., 2022)) | 69.90% | 58.80% |
| With Intervention ( (Hernandez et al., 2022)) | - | 63.70% |
| No Intervention (Ours) | 99.80% | 54.00% |
| Random Intervention (Ours) | 99.40% | 51.80% |
| Raw Attention | 98.20% | 81.40% |
| Raw Attention * Grad (Selvaraju et al., 2016) | 95.40% | 74.60% |
| Rollout Attention (Abnar & Zuidema, 2020) | 96.60% | 52.20% |
| Rollout Attention * Grad (Chefer et al., 2021) | 98.20% | 55.60% |
| Interpretation-based Intervention (Ours) | 99.20% | 88.80% |
| Interpretation-based Intervention w/ RS (Ours) | 99.20% | **89.20%** |

the intervention process is shown in Appendix D. If our intervention is successful, the transformer should predict airport through reasoning on concrete roads and airplanes.

**Reducing Spurious Correlations.** In contrast to work that modifies the model weights, we explore how our framework could remove spurious correlations through modifying token representations. We identify the tokens associated with spurious features and disable them with zero vectors. We will demonstrate our method's effectiveness for removing spurious features through classfication tasks that finetune a linear layer on CLS output embedding.

## 4 EXPERIMENT

**Dataset.** *VAW* dataset (Pham et al., 2021) is a large-scale visual attributes dataset with bounding box labels for the attribution annotation. We use it to study whether the annotated attribute emerges in vision transformer reasoning. We evaluate over the validation set, which contains 3297 images. *UC Merced Land Use Dataset* (Yang & Newsam, 2010) contains remote sensing satellite images of 21 classes, with 100 images in each class. We use it to study fixing typographical attacks and intervening in the visual reasoning procedure. We also use 2088 unique nouns and adjectives extracted from remote sensing image captions of RSICD dataset (Lu et al.) as textual vocabulary to retrieve interpretations from. *CelebA Dataset.* (Liu et al., 2015) We use the task of classifying the hair color as *gray* or *not gray*. The label is spuriously correlated with gender. Our goal is to use our framework to remove these spurious correlations.

**Implementation Details.** Our experiment focuses on the CLIP-B/32 model, while our method is general and can be used for any other transformer-based architecture. We use a single Titan RTX GPU with 24GB memory for our experiment. We found that applying small *random smoothing* noise to output of each layer when forward propagate without attentions improves performance of model control tasks. We reason that random smoothing might help address potential embedding space distribution shift resulting from disabling attention. Details are discussed in appendix B. We denote results obtained using random smoothing with *RS* in the following.

### 4.1 EVALUATING MODEL INTERPRETATION QUALITY

**Quantifying Token Interpretation Quality through Causal Intervention.** To quantitatively evaluate INViTE interpretation's quality, we employed causal tracing, an established method to interpret neural networks Meng et al. (2022). We will study whether our text explanation interprets

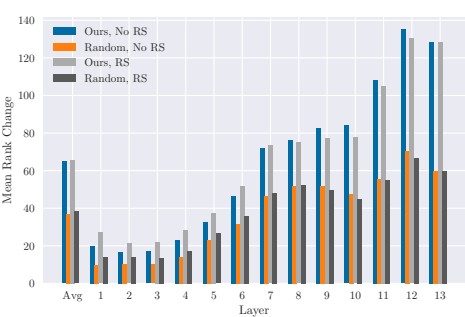

(a) Rank change under causal intervention on objects. Masking out interpreted objects result in larger rank change of interpreted descriptions than applying random masks, indicating that our interpretations are aligned with the actual object.

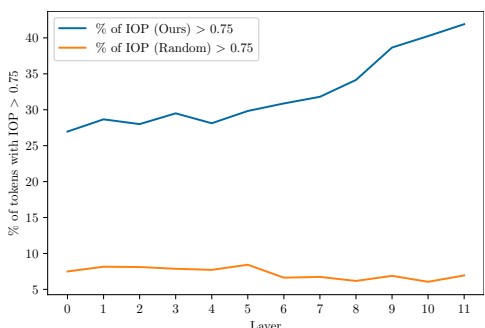

(b) Percentage of tokens having high overlap with VAW bounding box labels (IOP > 0.75) in different layers. Overlaps between bounding box labels and image areas most crucial to tokens interpreted as the labeled objects significantly exceed overlaps obtained with random masks, indicating our interpretation's effectiveness in capturing visual concepts.

Figure 4: Results from quantitative evaluations of interpretation quality.

the right object by masking out that object from the input. We first interpret each token with our method (top text in cosine correlation ranking). Then, we mask all presence of the object that the token interprets to with bounding boxes provided by VAW and produce another cosine correlation ranking of texts for each token. We then measure how much the original top-ranked text changes in ranking after the masking intervention. We focus on the tokens that interpret to object names or positive attributes labeled in VAW. A large rank change implies that our method provides meaningful interpretations since masking out the object directly affects our token explanation. We also compare this result to rank changes after applying random masks (same total area as VAW mask for the image) as a baseline.

Figure 4a shows the results of rank change after masking out the explained objects. For both vanilla interpretations and interpretations with random smoothing (RS), masking out the labeled objects causes interpretions' rank change larger than applying random masks by 40% to 120%. This significant gap demonstrates the validity of our text explanations through their tight links to the annotated attributes.

**Quantifying Token Interpretation Quality through Saliency Map Overlap.** Rollout attention flow (Abnar & Zuidema, 2020) provides a way to associate a token and image area it attends to. After min-max normalization of the rollout attention weights, we produce a mask with a threshold of 0.9 in the input space. Suppose a token interprets to be one of the object names in the image annotation, we calculate the Intersection over prediction (IOP) ratio between the produced mask and the union of all bounding boxes of the interpreted object. IOP is formally defined as follows: $IOP = \frac{\text{area}(truth \cap prediction)}{\text{area}(prediction)}$. Note that the denominator differs from the commonly used Intersection over Union method, which uses the union of truth and prediction as the denominator. We modify this because when a token's saliency mask only covers a part of an object, it is still a valid interpretation. For example, if a token is interpreted as a *car*, and its saliency map only covers a part of the car, the *car* is still a correct explanation for that highlighted part. We measure the percentage of tokens whose attended areas have high overlaps with ground truth bounding boxes (IOP > 0.75). We compare this percentage with that resulted from random masks with the same area as annotated bounding box area.

The result is shown in Figure 4b. The large difference between percentages obtained from ours and random masks demonstrates that our text explanation is better aligned with the ground-truth objects, reinforcing the effectiveness of our interpretation in capturing visual concepts.

### 4.2 Controlling Vision Transformer via Text Explanation

**Fixing Typographical Attacks.** Typographical attacks (Santurkar et al., 2021) are real-world threats for vision foundation models. We tested fixing typographical attacks with satellite images of forest class (100 images). The task zero-shot classifies between 5 classes *ocean*, *forest*, *runway*, *parking*,

Table 3: The effectiveness of our interpretation for model editing. We show the prediction from the original model, from random replacing, from using our approach, and from our approach with random smoothing (RS). Using our approach with random smoothing can provide better intervention to the reasoning. Our method achieves 77% of the intervention efficiency. This demonstrates that our token interpretation not only explains what each token represents but also the causal relationship between a token's and final image's representation in ViT's reasoning process.

|  | No Replace | Random Replace | Ours | Ours (RS) |
|---|---|---|---|---|
| Highway ↓ | 100.00% | 82.00% | 27.00% | **5.00%** |
| Airport ↑ | 0.00% | 18.00% | 73.00% | **95.00%** |

Table 4: Debias CelebA result. We show the classification accuracy of original linearly probed CLIP model (baseline) and our approach, using our interpretation guided intervention largely reduce the spurious correlations and improve model's performance on the worst group.

|  | Weighted Average ↑ | Male Gray Hair↑ | Male Non-Gray Hair↑ | Female Gray Hair↑ | Female Non-Gray Hair↑ |
|---|---|---|---|---|---|
| Baseline | 58.22% | **99.67%** | 15.85% | 19.68% | **99.67%** |
| Raw Attention | 70.00% | 90.71% | 42.00% | 59.50% | 88.92% |
| Raw Attention * Grad (Selvaraju et al., 2016) | 66.17% | 99.10% | 19.01% | 49.52% | 98.86% |
| Rollout Attention (Abnar & Zuidema, 2020) | 69.95% | 91.61% | 38.27% | 59.98% | 91.12% |
| Rollout Attention * Grad (Chefer et al., 2021) | 62.48% | 99.43% | 25.20% | 28.53% | 98.78% |
| Our Intervention | 81.66% | 97.23% | 74.01% | 58.00% | 98.29% |
| Our Intervention (RS) | **83.91%** | 97.23% | **74.80%** | **66.56%** | 97.80% |

and *residential* by selecting the top cosine correlation pair between CLS embedding and label text embedding. We conduct typographical attacks by overlaying random white boxes containing black text *ocean* on images. Example attack and how our interpretation reveals attack's effects on model reasoning are shown Figure 5. To fix typographical attack, we replace tokens that interpret to the following descriptions with zero vectors: *word*, *text*, *a word*, *a line of word*, etc. Note that our method does not require any assumption about the actual content of the attack word. To demonstrate the effectiveness of fixing attacks with our natural language interpretations, we also measure results obtained by randomly removing tokens (the number of randomly removed tokens equals the number of tokens interpreted to word-related descriptions).

Table 2 shows the result on original and typographically attacked images. Inferences with and without intervention all perform well on original images. Typographical attack subverts 95% of the images to the wrong prediction. If we remove random latent tokens by the same number as our intervention, it only mitigates 17% of the examples. Using our text explanation as guidance and removing the tokens related to words, we can correct 97% of the samples, demonstrating the accuracy of our text explanation and how it can help fix the typographical attacks. On average, our intervention (with RS) replaces $0.533$ tokens per layer.

To compare our method's effectiveness of fixing typographical attacks with previous works, we followed the experiment setup in Hernandez et al. (2022), which uses images from 10 categories (50 images per category) in ImageNet validation set. Typographical attack is conducted by overlaying the text of another randomly selected category's label on the image. The procedure of identifying and removing word related tokens resembles the setup above. While best improvements on attacked sample accuracy in Hernandez et al. (2022) is 4.9%, we improved accuracy on attacked images by 35.20%, demonstrating the effectiveness of our text explanation in finding the right token. We also compare editing based on our method and editing based on following saliency map based methods: Raw attention values from penultimate layer to final layer CLS token; GradCAM Selvaraju et al. (2016) with the same implementation for ViT in Chefer et al. (2021) (raw attention values from penultimate layer to final layer CLS tokens * their gradient respect to similarity with prompt *this can be described as a text*); Rollout attention from CLS token to input layer Abnar & Zuidema (2020); Rollout attention from CLS token to input layer where each attention matrix is multiplied by its gradient respect to similarity with prompt *this can be described as a text* Chefer et al. (2021). To remove typographical attack, we mask the parts of the image with map $>$ threshold with 0 and report results with best threshold. We found that our method outperforms all saliency map based methods.

**Object Entity Intervention.** On the UC Merced land use dataset, we pick the images that are classified as *Highway*, and test if our approach can modify the prediction to be *Airport* by intervening on the latent reasoning procedure. For our random replacing, we replace the same number of tokens as the number of tokens that interpret to associate with cars. Since features in different transformer

layers lie in different representation spaces, we replace tokens with those from the same layer. Results are shown in Table 3, demonstrating that our interpretations allow for effective model intervention. On average, our intervention (with RS) replaces 1.972 tokens per layer. Details on the layerwise number of replacements are shown in Appendix E.

**Remove Gender Bias.** Machine learning models often fail due to attending to spurious correlations. In the CelebA dataset, gender can be a spurious correlation for predicting hair color. We conduct a binary classification of *gray hair* vs. *non-gray hair* on the CelebA dataset by fine-tuning a linear layer on CLS token embedding. We fine-tune the linear layer for one epoch with Adam optimizer (lr = $10^{-3}$). We first fine-tune a model without intervention and measure accuracy for each class. Then, we fine-tune a model while replacing all tokens on layer 12 that interpret to descriptions outside a list of hair-related

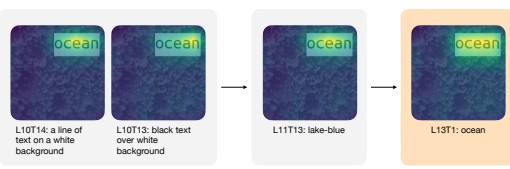

Figure 5: An example token interpretations in the presence of typographical attack. The model derives *lake-blue* from tokens that attend to text *ocean* and eventually produces false CLS token interpretation *ocean* instead of *forest*.

words (*hair*, *gray hair*, *gray*, *not gray hair*, *hairstyle*, *curl hair*, *straight hair*) with zero vector. At inference, we conduct the same replacement. In addition to the hair words, we use 5000 randomly sampled unique nouns and adjectives extracted from GPT description for ImageNet (Mao et al., 2022b) and words related to gender such as *gender*, *male*, and *female* (a full list is shown in Appendix C) as the vocabulary for interpretation. We also manually avoided the replacement of the CLS token to avoid excessive information loss. Table 4 shows the accuracy before and after our intervention to remove the gender bias. Our text explanation can guide the intervention and achieve up to 59% performance gain in the worst group, demonstrating the accuracy of our text explanation. We compare editing performance based on our interpretation to that based on the same saliency map based methods above. We mask out image parts with non-gradient based map > threshold or with gradient-based map < threshold with 0, where gradient is taken respect to respect to similarity with *this can be described as hair* and best threshold is reported. We found our method outperform all saliency maps. In addition, random smoothing can provide a better explanation, which leads to even higher overall performance. On average, we replaced 43.58 tokens on layer 12 (with RS). Note that this number is larger than the previous two experiments since our intervention aims to retain a minor detail and remove most other features.

### 4.3 ANALYSIS

**Our Interpretations Reveal Visual Reasoning.** In example latent token interpretations under our framework in Figure 3, we demonstrate our framework's capability to explain complex interactions in images, such as *the farm animals are eating grass and seem to be enjoying the farm*. In addition, our framework reveals the emerging visual reasoning procedure in the transformer. In the middle section, we show interpretations of tokens from layer $K$ that the tokens from $K + 1$ shown in right section attend to most and thus demonstrate vision transformer's hierarchical reasoning process. For example, the model combines latent tokens representing objects (*motor vehicle*) and context (*valley*) to predict an *overpass*. To recognize cows in a field, the model combines *black and white*, *legs*, and *grass*. We interpreted the first two images from the VAW dataset with vocabulary extracted from MILAN annotations (Hernandez et al., 2022). The second to fifth images from UC Merced Land Use Dataset are interpreted with vocabulary extracted from RSICD captions. The last image from the CelebA dataset is interpreted with vocabulary extracted from the GPT description ImageNet.

## 5 CONCLUSION

Our work proposes INViTE, a novel methodology for interpreting the latent tokens of visual foundation models with natural language descriptions by mapping token representations to the semantic space of the final layer with local transformer operations. Our approach eliminates the need for additional model training or data collection and extends to open-world and domain-specific datasets well. Qualitative and quantitative experiments demonstrate the effectiveness of our text interpretation. Our methodology gives insights on how visual foundation models reason to provide their predictions. Model editing based on our interpretations allow users to control model reasoning behaviors and enhance model robustness against spurious features and biases. Our method facilitates increased transparency and usability, fostering the development of accountable and responsible AI.

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

## A    USING OUR INTERPRETATION TO UNDERSTAND ADVERSARIAL ATTACK

Given image $X$ of class $c$, we conduct adversarial attack on CLIP by finding $\epsilon$ such that CLIP makes wrong zero-shot classification with $X + \epsilon$ as class $c'$. We find such $\epsilon$ through optimizing following loss through gradient descent:

$$\epsilon = \underset{\epsilon}{\text{argmin}} \, \text{CrossEntropy} \left( M(X + \epsilon)^T T(\text{text}), e_{c'} \right) + \lambda \|\epsilon\|_2$$

where $e_{c'}$ is a one-hot vector with $c'$-th location equals 1, $M$ is CLIP image encoder, $T(\text{text})$ is CLIP's encoding of zero-shot class prompts. We used regularization coefficient $\lambda = 0.01$.

We conduct such attack on an image of *forest* class as showin in 6. There's no visible difference for image before and after attack, but CLIP misclassifies 6b as *beach*.

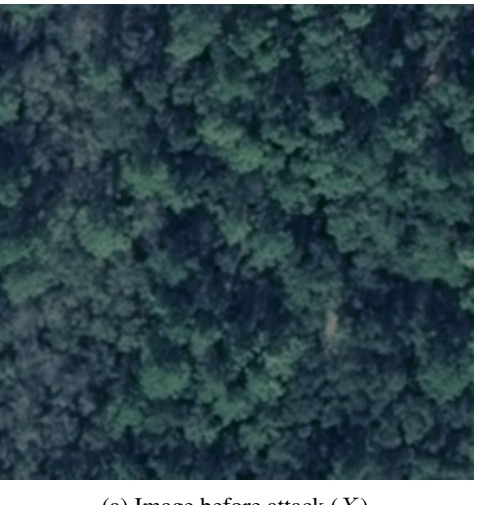 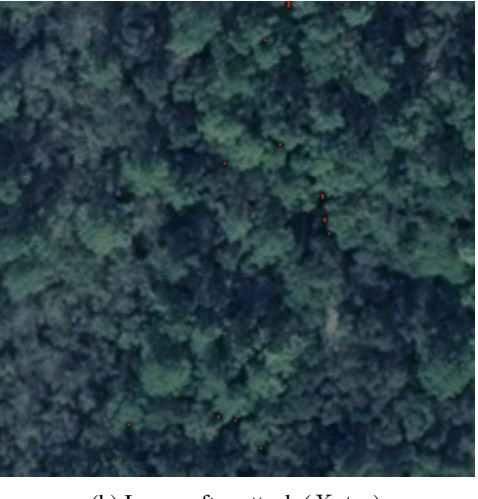

(a) Image before attack ($X$)          (b) Image after attack ($X + \epsilon$)

Figure 6: Comparison of forest images before and after the attack. There's no visible difference. CLIP zero-shot classifies 6a to *forest* and 6b to *beach*.

Table 5: Example interpretation of latent tokens before and after adversarial attack.

| Layer | Token | Before Attack | After Attack |
|-------|-------|---------------|--------------|
| 5 | 8 | vegetation | ultramarine |
| 12 | 7 | forest | rain |
| 12 | 8 | texture | snow |
| 12 | 11 | forest | pool |
| 12 | 15 | rattan | algae |

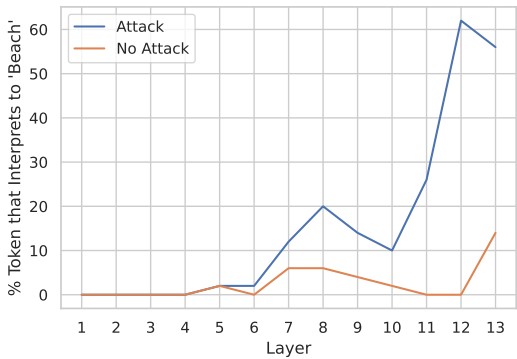

Figure 7: Percentage of tokens interpreting to *Forest* vs. *Beach* before and after adversarial attack on each layer. Adversarial attack impacts model most starting around layer 10.

To qualitatively examine effects of adversarial attack from latent tokens, we first interpret CLIP latent tokens before and after attack with open vocabulary. We provide a few examples in Table 5.

We also quantitatively study effects of adversarial attack. We interpret latent tokens with *This is forest* and *This is beach* and record the percentage of tokens that interpret to each prompt on every layer. Result is shown in Figure 7. Adversarial attack impacts model most starting around layer 10.

## B  MITIGATING DISTRIBUTION SHIFT IN TOKEN INTERPRETATION VIA RANDOM SMOOTHING

In model control experiments, we have found that adding random smoothing noise to each layer's output when forward propagate without attention increases model control performance. We reason that random smoothing might attain this improvement by addressing potential distribution shift resulting from disabling attention.

Our algorithm involves intervening in the original inference procedure of the transformer model and constraining it to perform computations on individual tokens. Once we disable the Multi-Head Self-Attention (MSA) operation, the calculations are limited to the MLP (Multi-Layer Perceptron) and BN (Batch Normalization) layers starting from the layer where token of interest situates. The MLP and BN layers move the token toward the final image-text representation space without additional training.

Since a transformer is trained with attention enabled, it achieves best performance when a token is combined with other tokens. Our method of disabling cross-attention creates a different distribution of token embeddings.

### B.1  DISTRIBUTION SHIFT ANALYSIS

To understand this distribution shift, we employed t-SNE (Van der Maaten & Hinton, 2008) analysis on the original and intervened token embedding distributions at different layers. Figure 9 shows that the token embeddings resulted from removing attentions largely reside in the same regions as those obtained from attentions enabled. The result also reveals that attention mechanism might plays a

Table 6: Average Drift

|                              | CLS     | Other   |
|------------------------------|---------|---------|
| CelebA                       | 0.08264 | 0.10314 |
| VAW                          | 0.07910 | 0.08703 |
| UC Merced Land Use (Forest)  | 0.09039 | 0.11492 |
| UC Merced Land Use (Highway) | 0.09020 | 0.10192 |

larger role in the penultimate layer to compose different concepts into high-level understandings as removing attentions create larger distribution shifts.

Over the VAW and CelebA dataset, we measure the distribution shift between with attention and without attention with $L_2$ norm between the center of the distributions. We obtain the $L_2$ difference between the mean vector of tokens produced from a previous layer with attention and that from a previous layer without attention. Note that we only measure the drift for layer 1 to layer 12 since interpreting layer 13 does not require disabling attention. In Table 6, we compared drift of CLS token and non-CLS token by averaging across layers. The result shows that CLS tokens results in smaller drifts than other tokens, agreeing with the histogram in Figure 8.

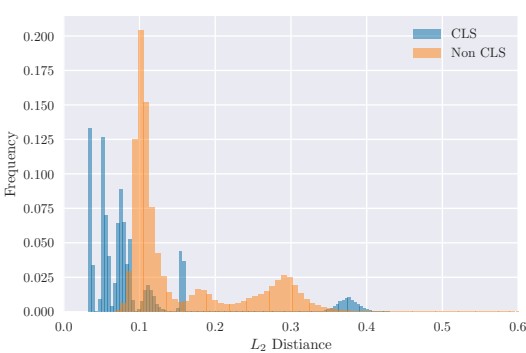

Figure 8: A histogram of $L_2$ distance of individual tokens with and without attention. We show the histogram of the shifts for CLS tokens and non-CLS tokens separately. CLS tokens tend to exhibits smaller drifts.

We acknowledge the distribution shifts introduced by removing attentions, especially for the penultimate layer. However, if the feed-forward model is robust to such distribution shifts and can extrapolate, accurate interpretations can still be obtained even with shifted embeddings. One crucial characteristic of a robust model under distribution shifting is smoothness, often indicated by a small Lipschitz constraint. For instance, linear models with regularization tend to extrapolate well out of the distribution and possess a small Lipschitz constraint. In contrast, deep models tend to have larger Lipschitz constraints and exhibit less smoothness. Since the operations from from layer 12 to the output layer is shallow and nearly linear, reasonable interpretations may still be retrieved despite the relatively larger gap observed before and after removing attentions.

### B.2 MITIGATING DISTRIBUTION SHIFT WITH RANDOM SMOOTHING

We propose incorporating random smoothing to further enhance the robustness of the feed-forward process against the distribution shift. Previous studies (Levine et al., 2019; Salman et al., 2019; Cohen et al., 2019) have shown that introducing random smoothing can reduce the Lipschitz constant and increase model smoothness. In our approach, we adopt random noise to smooth the latent tokens, mitigating the effects of distribution shift and maintaining the model's performance.

When applying random smoothing, for $j$-th token on layer $i$, we sample from a normal distribution with a mean of zero and standard deviation of the obtained average $L_2$ drift for $j$-th tokens on $i$-th layer, where each component is independent of each other. We then pass the latent tokens added with this noise through the layer 100 times and obtain the token for the next layer by taking an average.

## C VOCABULARY AND DETECTION WORDS

For fixing typographical attacks, we detect the text in latent representation by identifying tokens that interpret to following words: [*word*, *text*, *a word*, *a line of word*, *a line of text*, *a line of text on a white background*, *text over white background*, *black text*, *black text over white background*, *black alphabets*, *alphabets*, *letters*, *black letters*].

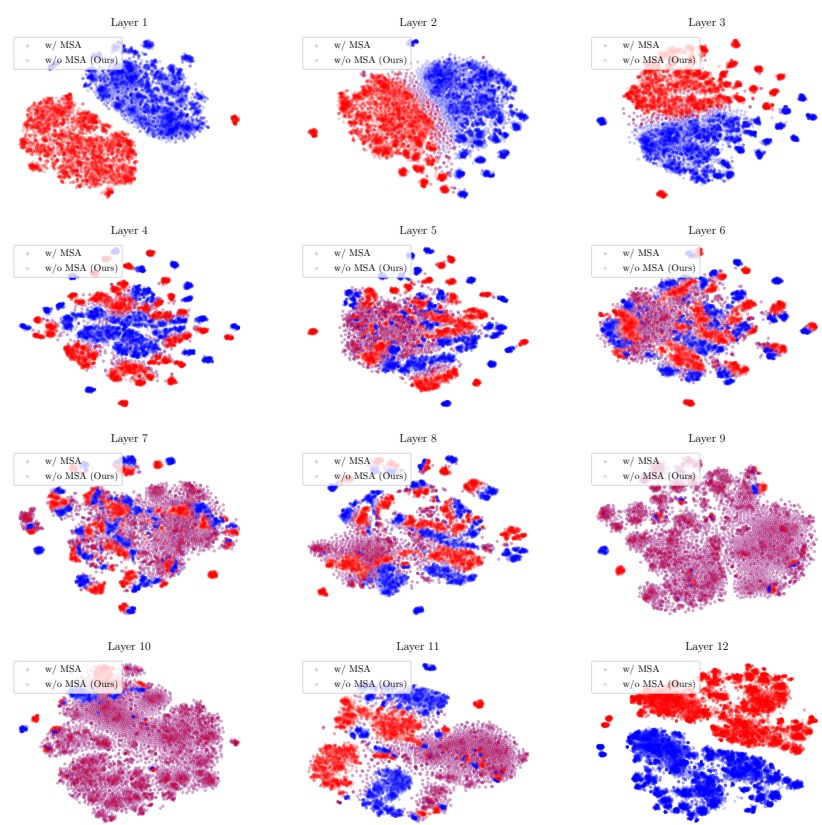

Figure 9: Distribution shifts introduced by removing cross attention. The blue/red dots are the original/our tokens. The purple color indicates they overlap in space. Token embeddings labeled as Ours on layer $i$ are obtained by disabling attention operations between layer $i - 1$ and $i$. This visualization is created from the CelebA dataset.

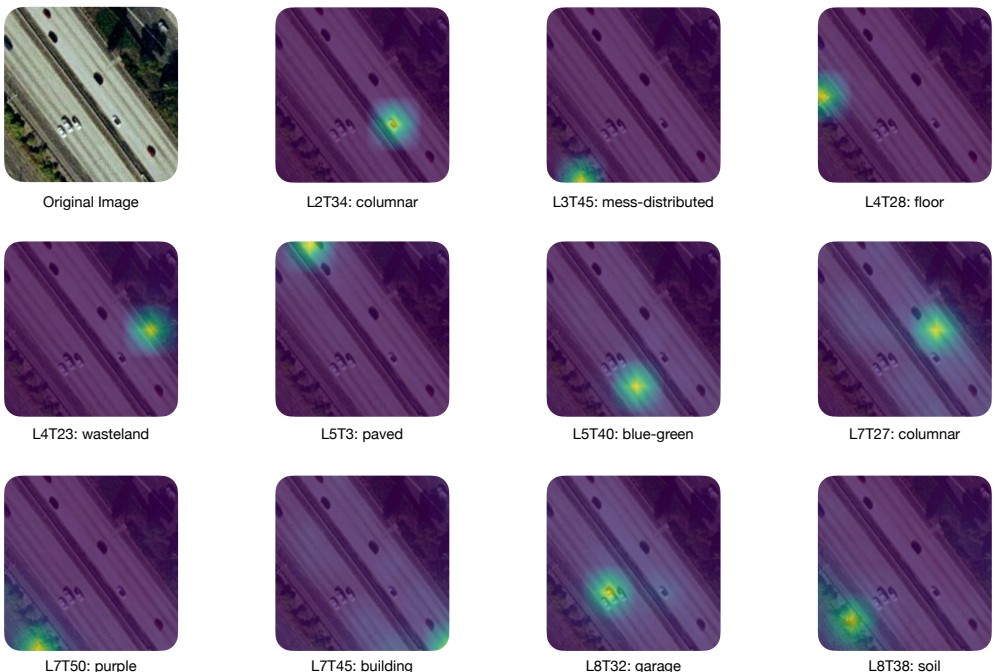

Figure 10: Example interpretations of tokens from earlier layers. Earlier layers often interpret to lower level features that reveal the model's reasoning process from more abstract concepts.

For intervening model reasoning experiment, we replace all tokens that interpret to car related tokens to airplane related tokens, which are randomly sampled from tokens that interpret to airplane in all airplane class images. Words used to detect airplane token: [*plane*, *aeroplane*, *aircraft*, *airplane*, *carrier*, *jet*, *airliner*, *flight*]. Words used to detect car tokens: [*car*, *automobile*, *motor vehicle*, *auto*, *truck*, *van*, *SUV*, *crossover*, *sedan*, *coupe*, *convertible*, *wagon*, *motorcycle*, *bike*, *bus*, *coach*, *lorry*, *trailer*, *caravan*, *camper*, *motorhome*, *RV*, *vehicles*].

For reducing the gender bias, we conduct classification of *gray hair* vs *non gray hair* on CelebA dataset through fine-tuning a linear layer on CLS token embedding. We fine-tuned the linear layer for one epoch with Adam optimizer (lr = $10^{-3}$). We first fine-tune a model without intervention and measure accuracy for each class. Then, we fine-tune a model while replacing all tokens on layer 12 that interpret to words other than a list of word related to hair (*hair*, *gray hair*, *gray*, *not gray hair*, *hairstyle*, *curl hair*, *straight hair*) with zero vector. At inference, the same replacement is done. In addition to the hair words, the vocabulary includes (1) 5000 randomly sampled unique nouns and adjectives extracted from GPT description for ImageNet; (2) words related to gender and features that model might use to decide gender: [*face*, *gender*, *male*, *female*, *man*, *men*, *woman*, *women*, *sex*, *boy*, *girl*, *man face*, *woman face*, *face*, *nose*, *eyes*, *breast*, *shoulder*, *mouth*, *ears*, *shoulders*, *shirt*, *jacket*, *tie*, *scarf*, *watch*, *hat*, *eyebrows*, *lashes*, *iris*, *pupils*, *nostrils*, *lips*, *teeth*, *tongue*, *temples*, *cheeks*, *cheekbones*, *chin*, *throat*, *arms*, *elbows*, *wrists*, *hands*, *fingers*, *chest*, *sternum*, *ribs*, *blouse*, *sweater*, *hoodie*, *blazer*, *pendant*, *watch*, *beanie*, *headband*]. We also manually avoided replacement of CLS token to avoid too significant of information loss.

# D   VISUALIZATIONS

We provide additional examples of our method's interpretation and associated token visualizations for earlier layers in models in Figure 10. In Figure 11, we show interpretations of a part of model on an airport image and a highway image. We also illustrate the procedure of extracting and replacing token in our intervention of model reasoning.

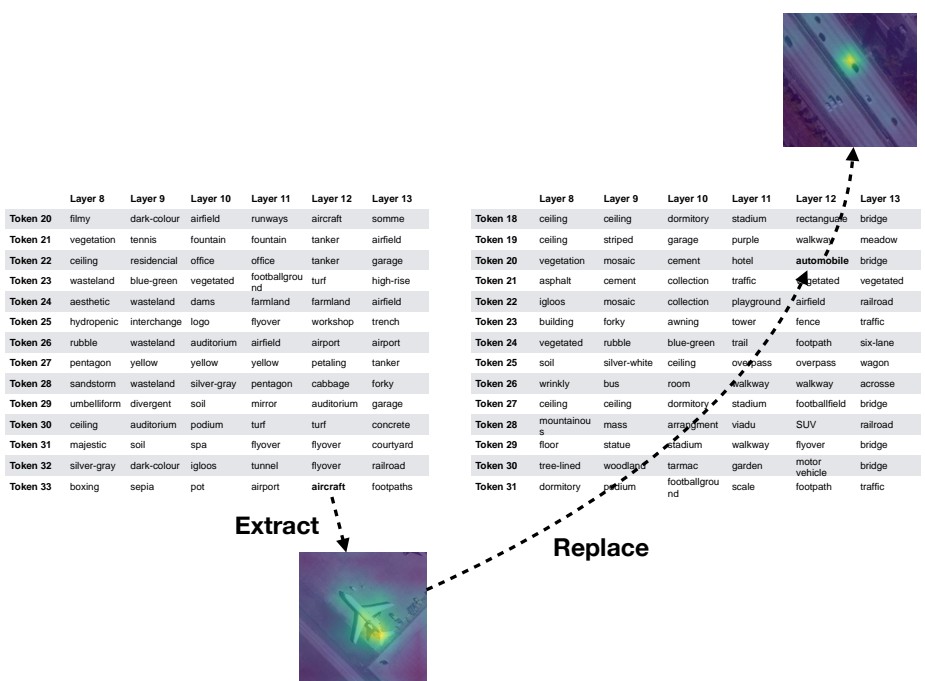

Figure 11: An example interpretations of a part of model and demonstration of intervening process on reasoning procedure by extracting *aircraft* token from the airport image and replacing the *automobile* token in the highway image. The model then combines the information of concrete road environment and airplane object to classify the image as *airport*.

Table 7: Average Number Tokens Replaced per Layer (RS). We show the average number of latent tokens that are interpreted to contain the semantic of interest. For the 50 tokens in each layer of our studied transformer, we find the tokens are replaced sparsely, but the higher layers tends to be replaced more. The task for debias is slightly different in nature than the first two tasks. In debias, we only hope to focus on one feature (hair color) and remove all other potentially spurious features, while in the first two tasks we hope to remove/edit small details and focus on information of the entire image. Therefore, the number of tokens replaced in the last task is much larger. To prevent over-damage of image features, we only removed tokens on the final layers (which contains best quality interpretations for object-level concepts and directly contribute to CLS) and left tokens on earlier layers intact.

| Layer | 1 | 2 | 3 | 4 | 5 | 6 | 7 | 8 | 9 | 10 | 11 | 12 |
|---|---|---|---|---|---|---|---|---|---|---|---|---|
| Typographical Attack | 0.04 | 1.04 | 0.24 | 0.27 | 0.52 | 0.46 | 0.65 | 0.76 | 0.73 | 0.53 | 0.67 | 0.48 |
| Entity Editing | 1.52 | 1.23 | 2.22 | 1.58 | 1.15 | 1.36 | 1.04 | 0.94 | 1.73 | 2.46 | 6.02 | 2.41 |
| Debias | 0 | 0 | 0 | 0 | 0 | 0 | 0 | 0 | 0 | 0 | 0 | 43.58 |

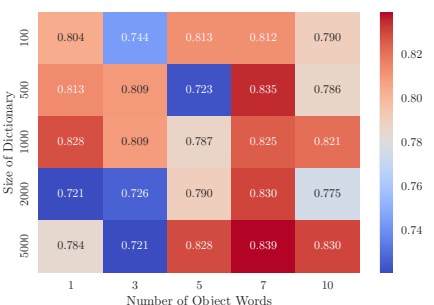

(a) Weighted Average Accuracy.

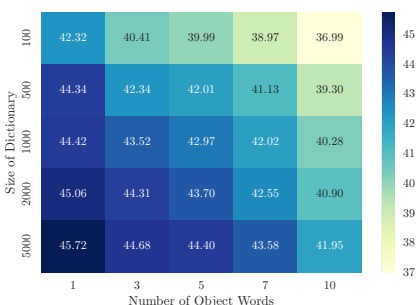

(b) Number of Tokens Replaced.

Figure 12: Ablation study on changing vocabulary/object word list size. (a) Balanced, larger vocabulary and object word list sizes improve performance. (b) Those sizes affect the number of tokens replaced.

## E  TOKEN REPLACEMENTS

In Table 7, we show the average number of tokens replaced on each layer in different experiments.

## F  EFFECTS OF VOCABULARY SIZE

Our approach produces natural language descriptions by retrieving the closest match from a set of provided vocabulary. During model editing, our method selects tokens that interpret to a list of object words of interest to determine which tokens to edit. We conduct an ablation study on the effect of vocabulary and object word list sizes over removing gender bias in CelebA dataset in Figure 12, where we find (1) larger vocabulary and object word list can generally improve performance; (2) performance is optimal when vocabulary and object word list sizes attain balance, so that information is not over-removed or under-removed as shown in 12b. These findings provide guidance on choosing appropriate vocabulary when using our framework for model editing.

