# OpenReview forum: "INViTE: INterpret and Control Vision-Language Models with Text Explanations"
_ICLR.cc/2024/Conference — ICLR 2024 poster_

### Official Review · Reviewer_ZFTq · 2023-10-27

**Soundness:** 3 good
**Presentation:** 3 good
**Contribution:** 2 fair
**Rating:** 5
**Confidence:** 4

**Summary:**

This paper proposes a novel method to interpret the latent tokens in pretrained vision-language models like CLIP using natural language descriptions. The key idea is to map the latent token embeddings to the final output space by disabling self-attention and propagating through the feedforward layers. This allows retrieving the closest text description for each token from the model's vocabulary. The authors demonstrate how these interpretations can provide insights into the model's reasoning and enable controlling model behaviors like fixing adversarial attacks, reducing biases, and replacing entities.

**Strengths:**

1. Novel method to interpret transformer tokens with language; doesn't require retraining or new data.

2. Solid motivation,clear methodology, extensive experiments across multiple datasets.

3. Well-written paper with clear explanation of the approach and results.

4. Interpretability is valuable for ML model transparency and trust. Controlling models via token editing has useful applications.

**Weaknesses:**

1. More analysis could be provided on how distribution shifts from removing attention affect interpretation quality.

2. More comparisons to related interpretation methods would further validate advantages.

3. Significance could be boosted by showing applications beyond the demonstrated tasks.

**Questions:**

Can you provide more details on how the distribution shift introduced by disabling attention affects interpretation quality? Is performance very sensitive to this?

What processes did you use to create the vocabularies for retrieving token interpretations? How important is vocabulary size and coverage?

How do your interpretation results compare qualitatively to other methods like saliency maps or concept activation vectors?

---

> ### Author Response · Authors · 2023-11-22
> **Thank you for your valuable reviews. We have addressed the concerns below.**
>
> **Analysis of distribution shift**
>
> Thank you for asking for clarification on the distribution shift. We provide numerical and visualization analysis of distribution shift in appendix B. We applied random smoothing to address the distribution shift (Levine et al., 2019; Salman et al., 2019; Cohen et al., 2019) and found random smoothing improves experimental results. The experiment performance with and without random smoothing therefore provides a proxy evaluation of distribution shift’s effects on interpretation quality. Most of our experiments (causal intervention in Fig. 4a; fixing typographical attack in table 1,2) show that distribution shift’s effects on interpretation are small. In a few other cases (entity editing in table 3; debias CelebA in table 4), addressing distribution shift with random smoothing improved the result more significantly.
>
> **Comparison to other interpretation methods**
>
> Thank you for asking for more comparison with other interpretation methods. To our knowledge, our method is among the first to provide token level interpretations of concepts. Previously methods mostly either interpret model weights or attribute prediction to image areas. Token level interpretation provides unique advantages of understanding better model reasoning processes and more flexible model editing. Therefore, it is difficult to directly compare our method with previous interpretation methods.
>
> In our paper, we use rollout attention saliency map to find the image region that is most influential on the token being interpreted. This provides a qualitative visual verification that our interpretation is correct, and allows us to quantitatively verify correctness through saliency overlap experiment.
>
> However, saliency map only provides correspondence between image area and token of interest. It cannot provide a conceptual explanation of a token. Although one could mask out images to keep only saliency map regions and find a text description of that region, this method would provide a same explanation for tokens that have similar saliency maps. However, our method is able to interpret tokens corresponding to the same image area differently. For example, in the fifth row of Figure 3, L12T50, L12T42, L13T1 all correspond to similar image areas, but interpretation is different. This allows for interpretations of different levels of concepts (e.g. “L12T35: green and blue coloured spots” contains low level features; “L13T1: a person walking down the street” contains high level concepts).
>
> We also conducted additional experiments that show our interpretation enabling better model editing in addressing typographical attacks and spurious correlations than other interpretation methods (see next response).

---

> ### Author Response · Authors · 2023-11-22
> **Response continued**
>
> **Quantitative experiment: Comparison with other interpretation methods**
>
> _Fixing Typographical Attacks_
>
> We consider the following methods of producing saliency maps.
>
> - Raw attention values from penultimate layer to final layer CLS token
> - GradCAM [1] with the same implementation for ViT in [3]: raw attention values from penultimate layer to final layer CLS tokens * their gradient respect to similarity with prompt “this can be described as a text”
> - Rollout attention from CLS token to input layer [2]
> - Rollout attention from CLS token to input layer where each attention matrix is multiplied by its gradient respect to similarity with prompt “this can be described as a text” [3]
>
> With all these saliency maps, we mask the parts of the image with map > threshold with 0. We test different thresholds and report best performance with each map.
>
> We perform the editing on the ImageNet typographical attack experiment (Table 2). Here’s the result:
>
> | Method                                  | Accuracy (original image) ↑ | Accuracy (attack image) ↑ |
> | --------------------------------------- | --------------------------- | ------------------------- |
> | Raw Attention                          | 98.20%                      | 81.40%                    |
> | Raw Attention * Grad [1] | 95.40%                  | 74.60%                    |
> | Rollout Attention [2] | 96.60%                     | 52.20%                    |
> | Rollout Attention * Grad [3] | 98.20%                 | 55.60%                    |
> | Ours | 99.20%                      | 88.80%                    |
> | Ours (w/ RS) | 99.20%                | **89.20%**                    |
>
> Our method outperforms the best saliency map based method (raw attention) by 7.8% on attack image accuracy.
>
> _Reducing spurious correlations_
>
> We conducted the removing spurious correlation experiment (Table 4) too with saliency maps. For raw attention and rollout attention, we mask out parts of the image with map > threshold with 0. For gradient based map, we take gradient respect to similarity to “this can be described as hair” and mask out parts of the image with map < threshold with 0. We report performance under the best threshold for each saliency map.
>
> | Method | Weighted Average ↑ | Male Gray Hair↑ | Male Non-Gray Hair↑ | Female Gray Hair↑ | Female Non-Gray Hair↑ |
> | --- | --- | --- | --- | --- | --- |
> | Baseline | 58.22% | 99.67% | 15.85% | 19.68% | 99.67% |
> | Raw Attention | 70.00% | 90.71% | 42.00% | 59.50% | 88.92% |
> | Raw Attention * Grad [1] | 66.17% | 99.10% | 19.01% | 49.52% | 98.86% |
> | Rollout Attention [2] | 69.95% | 91.61% | 38.27% | 59.98% | 91.12% |
> | Rollout Attention * Grad [3]| 62.48% | 99.43% | 25.20% | 28.53% | 98.78% |
> | Our Intervention | 81.66% | 97.23% | 74.01% | 58.00% | 98.29% |
> | Our Intervention (RS) | 83.91% | 97.23% | **74.80%** | **66.56%** | 97.80% |
>
> Our method outperforms best saliency map based method (raw attention) by 24.56% on worst class accuracy.
>
> [1] Selvaraju, R. R., Cogswell, M., Das, A., Vedantam, R., Parikh, D., & Batra, D. (2016). Grad-CAM: Visual Explanations from Deep Networks via Gradient-based Localization. arXiv preprint arXiv:1610.02391.
>
> [2] Samira Abnar and Willem Zuidema. Quantifying attention flow in transformers. In Proceedings of the 58th Annual Meeting of the Association for Computational Linguistics, pp. 4190–4197, Online, July 2020. Association for Computational Linguistics. doi: 10.18653/v1/2020.acl-main.385. URL https://aclanthology.org/2020.acl-main.385.
>
> [3] Hila Chefer, Shir Gur, and Lior Wolf. Transformer interpretability beyond attention visualization. In Proceedings of the IEEE/CVF Conference on Computer Vision and Pattern Recognition (CVPR), pp. 782–791, June 2021.

---

> ### Author Response · Authors · 2023-11-22
> **Response continued**
>
> **Effects of vocabulary**
>
> One advantage of our framework is that it could produce interpretation with any vocabulary of any type and size. In general, a larger and diverse vocabulary set provides more granular and comprehensive interpretations. We conducted an ablation study on how vocabulary size influences model editing results based on interpretations by measuring performance on reducing CelebA spurious correlation (Appendix F, Fig. 12). In general, a larger vocabulary (we tested up to 5000) performs better, but a balance between vocabulary and object word list size is important.
>
> Choosing specialized vocabulary might help improve interpretations on domain-specific images. For example, we used 2088 unique nouns and adjectives extracted from remote sensing image captions of RSICD dataset (Lu et al.) to interpret satellite images.
>
>
> **Process of vocabulary creation**
>
> To interpret VAW, CelebA, ImageNet images, we used vocabulary extracted from MILAN annotations (Hernandez et al., 2022) and GPT description of ImageNet (Mao et al., 2022b) which contain descriptions of most daily life object attributes. To interpret satellite images, we used 2088 unique nouns and adjectives extracted from remote sensing image captions of RSICD dataset (Lu et al.).
>
>
> **Applications beyond demonstrated tasks**
>
> Beyond the demonstrated tasks, our method could be generalized to pinpoint the internal mechanism for any kind of model errors. One can then build models that emphasize self-consistency of internal concepts and avoid overemphasis on certain biased features.
>
> We also conducted additional demonstrations to show that our interpretation allows for understanding how adversarial attacks impact the model in Appendix A. We show that adversarial attack impacts model the most starting from layer 10 and provide some example latent token interpretations before and after attack.

---

> > ### Comment · Reviewer_ZFTq · 2023-11-23
> > **Official Comment by Reviewer ZFTq**
> >
> > Thank you for the responses.  They provide clarification on the questions raised.  After discussing with other reviewers, I will reassess my score.

---

### Official Review · Reviewer_aYwS · 2023-10-27

**Soundness:** 3 good
**Presentation:** 2 fair
**Contribution:** 2 fair
**Rating:** 5
**Confidence:** 4

**Summary:**

The paper proposes a method to interpret the semantic meaning of vision foundation models by retrieving the closest text description. Specifically, the method turns off the inter-token cross attentions in the latter layers and only keeps the self-attention, in order to get the final CLS representation that corresponds to only the target token. Saliency attention visualizations and quantitative results are provided to show the correctness of the interpretations. To demonstrate the application of the proposed method in terms of controlling and intervening the model behaviors, the paper runs experiments on three tasks: fixing topographical attacks, intervening object entities, and removing gender bias.

**Strengths:**

1. The paper runs experiments on three practical tasks to showcase the applications of the derived interpretations. It is exciting to see that the interpretations can improve the model’s robustness towards text/topographical attacks and gender bias, as well as intervening the model’s decision in object classification in satellite images.

2. Abundant visualizations and numbers are reported to show the advantage of the proposed method over random intervention.

**Weaknesses:**

1. The major limitation is that the method can only be applied on the CLIP model (at least only CLIP is shown in the paper). Since CLIP aligns the visual embeddings (CLS) with the texts, the method can retrieve texts based on the CLS embedding after turning off the cross-attentions. Otherwise, if the model’s token embeddings are not trained to be aligned with the texts, it is doubtful whether the method can still work or not. It would be best if the authors can show that the method can workin on other non-text trained models like MAE, DINOv2, etc.

2. Most of the comparisons are against the “random baseline”, which is not a very strong baseline. For example, additional stronger baselines should be compared with, like the ones listed in the “model interpretation” in the related works sections. Moreover, it’s better to include related work [1].

3. The method is based on text-retrieval, which assumes a finite set of candidate texts (closed world). In related work, the authors criticize that Koh et al (2020) (concept bottleneck network) is limited to a closed vocabulary, but isn’t this work also closed vocabulary?

4. The method is training free. As an extension, can the method be extended for improving model training?

[1] (arxiv 2301.13081 STAIR: Learning Sparse Text and Image Representation in Grounded Tokens)

**Questions:**

See weakness.

---

> ### Author Response · Authors · 2023-11-22
> **Thank you for your valuable reviews. We have addressed the concerns below.**
>
> We thank the reviewer for their valuable comments. We are glad that the reviewer found our work to be exciting. We addressed the questions below:
>
> **Interpreting Non-Text Trained Models**
>
> Thank you for pointing this out. Our framework is general to all transformer architecture. For non-text trained models like DINOv2 and MAE, one can train a linear mapping from final CLS embedding (if available) or average pooling of all final layer embeddings to CLIP image-text embedding space. Then, one can interpret latent tokens in these models with our method.
>
> **Comparison to other interpretation methods**
>
> Thank you for pointing us to STAIR, which is an interesting extension of CLIP that provides better concept representation through sparse token space. We have revised our paper and included it in related work.
>
> Thank you for asking for more baselines. To our knowledge, our method is among the first to provide token level interpretations of concepts. Previously methods mostly either interpret model weights or attribute prediction to image areas. Token level interpretation provides unique advantages of understanding better model reasoning processes and more flexible model editing. Therefore, it is difficult to directly compare our method with previous interpretation methods.
>
> However, we conducted additional experiments to compare the effectiveness of addressing typographical attacks and spurious correlation between our interpretation and saliency mask based interpretation methods (see next response).

---

> ### Author Response · Authors · 2023-11-22
> **Response continued**
>
> **Quantitative experiment: Comparison with previous interpretation method**
>
> _Fixing Typographical Attacks_
>
> We consider the following methods of producing saliency maps.
>
> - Raw attention values from penultimate layer to final layer CLS token
> - GradCAM [1] with the same implementation for ViT in [3]: raw attention values from penultimate layer to final layer CLS tokens * their gradient respect to similarity with prompt “this can be described as a text”
> - Rollout attention from CLS token to input layer [2]
> - Rollout attention from CLS token to input layer where each attention matrix is multiplied by its gradient respect to similarity with prompt “this can be described as a text” [3]
>
> With all these saliency maps, we mask the parts of the image with map > threshold with 0. We test different thresholds and report best performance with each map.
>
> We perform the editing on the ImageNet typographical attack experiment (Table 2). Here’s the result:
>
> | Method                                  | Accuracy (original image) ↑ | Accuracy (attack image) ↑ |
> | --------------------------------------- | --------------------------- | ------------------------- |
> | Raw Attention                          | 98.20%                      | 81.40%                    |
> | Raw Attention * Grad [1] | 95.40%                  | 74.60%                    |
> | Rollout Attention [2] | 96.60%                     | 52.20%                    |
> | Rollout Attention * Grad [3] | 98.20%                 | 55.60%                    |
> | Ours | 99.20%                      | 88.80%                    |
> | Ours (w/ RS) | 99.20%                | **89.20%**                    |
>
> Our method outperforms the best saliency map based method (raw attention) by 7.8% on attack image accuracy.
>
> _Reducing spurious correlations_
>
> We conducted the removing spurious correlation experiment (Table 4) too with saliency maps. For raw attention and rollout attention, we mask out parts of the image with map > threshold with 0. For gradient based map, we take gradient respect to similarity to “this can be described as hair” and mask out parts of the image with map < threshold with 0. We report performance under the best threshold for each saliency map.
>
> | Method | Weighted Average ↑ | Male Gray Hair↑ | Male Non-Gray Hair↑ | Female Gray Hair↑ | Female Non-Gray Hair↑ |
> | --- | --- | --- | --- | --- | --- |
> | Baseline | 58.22% | 99.67% | 15.85% | 19.68% | 99.67% |
> | Raw Attention | 70.00% | 90.71% | 42.00% | 59.50% | 88.92% |
> | Raw Attention * Grad [1] | 66.17% | 99.10% | 19.01% | 49.52% | 98.86% |
> | Rollout Attention [2] | 69.95% | 91.61% | 38.27% | 59.98% | 91.12% |
> | Rollout Attention * Grad [3]| 62.48% | 99.43% | 25.20% | 28.53% | 98.78% |
> | Our Intervention | 81.66% | 97.23% | 74.01% | 58.00% | 98.29% |
> | Our Intervention (RS) | 83.91% | 97.23% | **74.80%** | **66.56%** | 97.80% |
>
> Our method outperforms best saliency map based method (raw attention) by 24.56% on worst class accuracy.
>
> [1] Selvaraju, R. R., Cogswell, M., Das, A., Vedantam, R., Parikh, D., & Batra, D. (2016). Grad-CAM: Visual Explanations from Deep Networks via Gradient-based Localization. arXiv preprint arXiv:1610.02391.
>
> [2] Samira Abnar and Willem Zuidema. Quantifying attention flow in transformers. In Proceedings of the 58th Annual Meeting of the Association for Computational Linguistics, pp. 4190–4197, Online, July 2020. Association for Computational Linguistics. doi: 10.18653/v1/2020.acl-main.385. URL https://aclanthology.org/2020.acl-main.385.
>
> [3] Hila Chefer, Shir Gur, and Lior Wolf. Transformer interpretability beyond attention visualization. In Proceedings of the IEEE/CVF Conference on Computer Vision and Pattern Recognition (CVPR), pp. 782–791, June 2021.

---

> ### Author Response · Authors · 2023-11-22
> **Response continued**
>
> **Openness of vocabulary**
>
> Our approach works with vocabulary of any type and size, which makes our approach work on open vocabulary. In contrast, Koh et al (2020)’s method interprets a small set of predefined concepts, where the concept set is built into model architecture that is task-specific.
>
> For our method, we showed that large vocabulary (5000 descriptions) is beneficial for producing better descriptions in our framework in section 4.3. Our method also works on any type of vocabulary, which makes it easily adapt to different tasks. To enable better interpretations in domain specific tasks, one can simply choose a vocabulary set that contains domain-specific languages (for example, we used RSICD captions to interpret satellite images in UC Merced Land Use Dataset).
>
> **Improving model training with our interpretations**
>
> Thank you for your suggestion on extending our method to improve model training. Our approach can indeed be generalized to enhance model training. A few possible ways include: 1) regularize the model to emphasize self-consistency of internal concept representations for more concrete visual reasoning and enhanced robustness against adversarial attacks (we demonstrate how our method help understand adversarial attacks in Appendix A); 2) increase model robustness by steering the model to avoid overemphasis on certain classes of features.

---

### Official Review · Reviewer_jZ36 · 2023-11-01

**Soundness:** 2 fair
**Presentation:** 4 excellent
**Contribution:** 3 good
**Rating:** 8
**Confidence:** 4

**Summary:**

This paper introduces a method to interpret intermediate representations (referred to as *latent tokens*) in vision-language Transformer models (e.g. CLIP) by retrieving relevant natural language descriptions for individual latent tokens. The authors do so by removing the self-attention operations after a latent token, and using the last layer token representation to retrieve a text description. The authors posit that the retrieved text for a latent token provides an interpretation for it. They further posit that manipulating these latent tokens can be used to edit and control model behavior.

**Strengths:**

- The experiments are very convincing. The author use their methodology to not only do model interpretation (which would have been sufficient imo), but also for model controllability. The results for interpretability (causality and saliency map overlap) are very convincing.

- The controllability experiments are also very well designed, and demonstrate consistent results across three different kinds of model editing -- typography attacks, entity editing and gender debiasing.

- The paper is well-written overall. Some of the experiment setups are a little hard to understand, because the setting is slightly artificial,

**Weaknesses:**

- I am skeptical of the motivation behind the methodology. Specifically, typically the CLS representation from the vision encoder is used as the query to do text retrieval in CLIP, but do we know that latent tokens corresponding to other image patches from the final layer retrieve meaningful text concepts? There is no theoretical motivation -- which isn't strictly needed, but I would be much likelier to trust the method beyond just the empirical results.

- It's unclear what the benefit of these natural language descriptions for latent tokens is, or how these descriptions should be leveraged. The examples in Section 4.3 ("Our Interpretations Reveal Visual Reasoning") do not seem very convincing, and are more about how one chooses to interpret the tokens' NL descriptions (e.g. I would not think "motor vehicle" + "valley" = "overpass" is necessarily a correct reasoning, let alone whether that's actually how the model went about the reasoning process).

**Questions:**

- What is the benefit of the proposed method over the saliency maps that are presented in Figure 3?

---

> ### Author Response · Authors · 2023-11-22
> **Thank you for your valuable reviews. We have addressed the concerns below.**
>
> Thank you for your thoughtful review. We are glad that the reviewer found our experiments well-designed and convincing. We address the reviewer’s concerns below:
>
> **Motivation behind our methodology**
>
> Thank you for your suggestion on theoretical motivation.
>
> Let $x_{cls}$ be the CLS embedding after final projection layer. Let $x_i$ be the $i$-th non-CLS embedding after the final projection layer. Let $v_j$ be the embedding of $j$-th text description.
>
> We know that we can retrieve $\hat{j}$-th text as a meaningful text description for $x_{cls}$ by taking $\hat{j}= \arg\max_{j} x_{cls}^T v_j$.
>
> Now consider the average of all non-CLS embedding: $x_{agg} = \frac{1}{N}\sum_{i=1}^N x_i$. This average aggregates information from all image patches. We will empirically show that we could retrieve accurate text descriptions from averaged non-CLS tokens by showing $\arg\max_{j} x_{cls}^T v_j = \arg\max_{j} x_{agg}^T v_j$.
>
> Since the averaged token is a linear combination of each non-CLS token, $x_{agg}^T v_j =  \frac{1}{N}\sum_{i=1}^N x_i^T v_j$, being able to retrieve correct text descriptions for $x_{agg}$ implies that we could retrieve correct text descriptions for each individual non-CLS token $x_i$.
>
> We conducted a two-class zero-shot classification using CLS token and average of non-CLS tokens. CLS token embedding attains 99.5% accuracy; averaging non-CLS token embeddings on the final layer attains 97.5% accuracy. This experiment shows that indeed $\arg\max_{j} x_{cls}^T v_j = \arg\max_{j} x_{agg}^T v_j$, and non-CLS tokens also encode correct information for text retrieval. (The experiment is conducted on 100 images from each class from UC Merced Land Use Dataset with beach class and forest class).
>
> **Benefits of natural language descriptions**
>
> The text explanations produced from our method reveal information encoded in each latent token. This could help users easily pinpoint where do models make errors or produce bias, and users could address the issues with model editing guided by the interpretations (as we showed in fixing typographical attack and reducing spurious correlation experiments.)
>
> **Using our interpretations to understand visual reasoning**
>
> Thank you for mentioning our usage of text descriptions to understand visual reasoning. Using text descriptions of latent tokens to understand the model reasoning process in for example “motor vehicle + valley = overpass” is not a discretionary interpretation but grounded in both text descriptions and model mechanics. In figure 3, referenced by section 4.3, we show interpretation of tokens on layer K+1 and tokens on layer K that the K+1 layer tokens pays most attention to (based on attention weights). For example, “overpass” pays most attention to “motor vehicle” and “valley,” so we reason that this combination of local features into global features allows the model to understand the entire image as “overpass.”
>
> We agree that attention value alone doesn’t fully demonstrate causal relationship in reasoning. Therefore, we also conduct the Object Entity Intervention experiment in section 4.2. By showing that replacing car tokens with airplane tokens changes a model's understanding of the image from highway to airport, we demonstrate that our interpretation of local concepts are indeed causally related to how model reasons about global concepts.

---

> ### Author Response · Authors · 2023-11-22
> **Response continued**
>
> **Benefits over saliency map**
>
> In our paper, we use rollout attention saliency map to find the image region that is most influential on the token being interpreted. This provides a qualitative visual verification that our interpretation is correct, and allows us to quantitatively verify correctness through saliency overlap experiment.
>
> However, saliency map only provides correspondence between image area and token of interest. It cannot provide a conceptual explanation of a token. Although one could mask out images to keep only saliency map regions and find a text description of that region, this method would provide a same explanation for tokens that have similar saliency maps. However, our method is able to interpret tokens corresponding to the same image area differently. For example, in the fifth row of Figure 3, L12T50, L12T42, L13T1 all correspond to similar image areas, but interpretation is different. This allows for interpretations of different levels of concepts (e.g. “L12T35: green and blue coloured spots” contains low level features; “L13T1: a person walking down the street” contains high level concepts).
>
> We also conducted additional demonstrations to show that our interpretation allows for understanding how adversarial attacks impact the model in Appendix A. We show that adversarial attack impacts model the most starting from layer 10 and provide some example latent token interpretations before and after attack. Salient regions are unable to provide such understanding because there's no visible difference of image before and after adding adversarial attack.

---

> ### Author Response · Authors · 2023-11-22
> **Response continued**
>
> **Quantitative experiments comparing to saliency map based editing**
>
> We also conducted additional experiments that show our interpretation enabling better model editing in addressing typographical attacks and spurious correlations than saliency-map based methods:
>
> _Fixing Typographical Attacks_
>
> We consider the following methods of producing saliency maps.
>
> - Raw attention values from penultimate layer to final layer CLS token
> - GradCAM [1] with the same implementation for ViT in [3]: raw attention values from penultimate layer to final layer CLS tokens * their gradient respect to similarity with prompt “this can be described as a text”
> - Rollout attention from CLS token to input layer [2]
> - Rollout attention from CLS token to input layer where each attention matrix is multiplied by its gradient respect to similarity with prompt “this can be described as a text” [3]
>
> With all these saliency maps, we mask the parts of the image with map > threshold with 0. We test different thresholds and report best performance with each map.
>
> We perform the editing on the ImageNet typographical attack experiment (Table 2). Here’s the result:
>
> | Method                                  | Accuracy (original image) ↑ | Accuracy (attack image) ↑ |
> | --------------------------------------- | --------------------------- | ------------------------- |
> | Raw Attention                          | 98.20%                      | 81.40%                    |
> | Raw Attention * Grad [1] | 95.40%                  | 74.60%                    |
> | Rollout Attention [2] | 96.60%                     | 52.20%                    |
> | Rollout Attention * Grad [3] | 98.20%                 | 55.60%                    |
> | Ours | 99.20%                      | 88.80%                    |
> | Ours (w/ RS) | 99.20%                | **89.20%**                    |
>
> Our method outperforms the best saliency map based method (raw attention) by 7.8% on attack image accuracy.
>
> _Reducing spurious correlations_
>
> We conducted the removing spurious correlation experiment (Table 4) too with saliency maps. For raw attention and rollout attention, we mask out parts of the image with map > threshold with 0. For gradient based map, we take gradient respect to similarity to “this can be described as hair” and mask out parts of the image with map < threshold with 0. We report performance under the best threshold for each saliency map.
>
> | Method | Weighted Average ↑ | Male Gray Hair↑ | Male Non-Gray Hair↑ | Female Gray Hair↑ | Female Non-Gray Hair↑ |
> | --- | --- | --- | --- | --- | --- |
> | Baseline | 58.22% | 99.67% | 15.85% | 19.68% | 99.67% |
> | Raw Attention | 70.00% | 90.71% | 42.00% | 59.50% | 88.92% |
> | Raw Attention * Grad [1] | 66.17% | 99.10% | 19.01% | 49.52% | 98.86% |
> | Rollout Attention [2] | 69.95% | 91.61% | 38.27% | 59.98% | 91.12% |
> | Rollout Attention * Grad [3]| 62.48% | 99.43% | 25.20% | 28.53% | 98.78% |
> | Our Intervention | 81.66% | 97.23% | 74.01% | 58.00% | 98.29% |
> | Our Intervention (RS) | 83.91% | 97.23% | **74.80%** | **66.56%** | 97.80% |
>
> Our method outperforms best saliency map based method (raw attention) by 24.56% on worst class accuracy.
>
> [1] Selvaraju, R. R., Cogswell, M., Das, A., Vedantam, R., Parikh, D., & Batra, D. (2016). Grad-CAM: Visual Explanations from Deep Networks via Gradient-based Localization. arXiv preprint arXiv:1610.02391.
>
> [2] Samira Abnar and Willem Zuidema. Quantifying attention flow in transformers. In Proceedings of the 58th Annual Meeting of the Association for Computational Linguistics, pp. 4190–4197, Online, July 2020. Association for Computational Linguistics. doi: 10.18653/v1/2020.acl-main.385. URL https://aclanthology.org/2020.acl-main.385.
>
> [3] Hila Chefer, Shir Gur, and Lior Wolf. Transformer interpretability beyond attention visualization. In Proceedings of the IEEE/CVF Conference on Computer Vision and Pattern Recognition (CVPR), pp. 782–791, June 2021.

---

### Official Review · Reviewer_jxBW · 2023-11-01

**Soundness:** 2 fair
**Presentation:** 3 good
**Contribution:** 2 fair
**Rating:** 3
**Confidence:** 3

**Summary:**

The paper proposes a method to interpret the “latent” representations of vision transformers. To interpret a visual token representation of layer K, the paper proposes to disable the self-attention for layers higher than layer K, and then use the final layer representation to calculate text-token similarity. The ability to find such latent tokens enables several applications: fixing typographical attacks, intervening in the reasoning procedure, and reducing spurious correlations.

**Strengths:**

Such an interpretation approach is definitely new; it is a novel observation that one could simply disable the self-attention from above a certain layer (K) and then use the representation as a representation for latent token at layer K. The paper uses causal intervention and saliency map overlap to verify the effectiveness of the approach.

**Weaknesses:**

My main concern is that I do not quite see the method’s advantage compared to gradient-based methods that find important input regions (e.g., Grad-CAM) given a text description.

(1). The first question is why we want to find latent tokens but not salient regions?

Conceptually, the biggest advantage is that there might exist “high-level” and “abstract” latent tokens. For example, in Figure 3, using Grad-CAM to find regions corresponding to “overpass” might result in a lot of matched image patches while using the proposed method can find one single latent token corresponding to the concept.

However, this is not reflected in the quantitative experiments. For the fixing typographical attacks and reducing spurious correlations experiments, ideally a gradient-based baseline could be included, where we seek to use grad-cam to find and zero-out a few important image patches. Then one could compare whether the proposed method can achieve the same performance but zeroing out less tokens. Otherwise, it is hard to claim that the method can find “high-level” latent tokens.


(2). Suppose we wish to find latent tokens corresponding to a text description, why should we resort to the proposed method but not a gradient-based method where we calculate the gradient with respect to each latent tokens of every layer? This seems like a more principled way to obtain important latent tokens.

**Questions:**

I am not sure I get the motivation behind the Intervening in the Reasoning Procedure experiment. Under what settings would this be useful?

---

> ### Author Response · Authors · 2023-11-22
> **Thank you for your valuable reviews. We have addressed the concerns below.**
>
> We are glad that the reviewer found our method to be novel. We ran the suggested experiments and addressed the concerns below.
>
> **Advantage of text descriptions of latent tokens over explaining with salient regions**
>
> When interpreting latent tokens with salient regions, if different latent tokens correspond to the same salient region, the resulting explanation would be the same. However, our method is able to interpret tokens corresponding to the same image area differently. For example, in the fifth row of Figure 3, L12T50, L12T42, L13T1 all correspond to similar image areas, but interpretation is different. This allows for interpretations of different levels of concepts (e.g. “L12T35: green and blue coloured spots” contains low level features; “L13T1: a person walking down the street” contains high level concepts).
>
> We also conducted additional demonstrations to show that our interpretation allows for understanding how adversarial attacks impact the model in Appendix A. We show that adversarial attack impacts model the most starting from layer 10 and provide some example latent token interpretations before and after attack. Salient regions are unable to provide such understanding because there's no visible difference of image before and after adding adversarial attack.
>
> We also conducted additional quantitative experiments to show that text descriptions of latent tokens with our method produce better model editing than saliency map based editing (see next response).

---

> ### Author Response · Authors · 2023-11-22
> **Response continued**
>
> **Quantitative experiment: Comparison with gradient-based method**
>
> Thank you for asking for more baseline comparison. We conducted the following additional experiments to compare our methods against saliency based methods.
>
> _Fixing Typographical Attacks_
>
> We consider the following methods of producing saliency maps.
>
> - Raw attention values from penultimate layer to final layer CLS token
> - GradCAM [1] with the same implementation for ViT in [3]: raw attention values from penultimate layer to final layer CLS tokens * their gradient respect to similarity with prompt “this can be described as a text”
> - Rollout attention from CLS token to input layer [2]
> - Rollout attention from CLS token to input layer where each attention matrix is multiplied by its gradient respect to similarity with prompt “this can be described as a text” [3]
>
> With all these saliency maps, we mask the parts of the image with map > threshold with 0. We test different thresholds and report best performance with each map.
>
> We perform the editing on the ImageNet typographical attack experiment (Table 2). Here’s the result:
>
> | Method                                  | Accuracy (original image) ↑ | Accuracy (attack image) ↑ |
> | --------------------------------------- | --------------------------- | ------------------------- |
> | Raw Attention                          | 98.20%                      | 81.40%                    |
> | Raw Attention * Grad [1] | 95.40%                  | 74.60%                    |
> | Rollout Attention [2] | 96.60%                     | 52.20%                    |
> | Rollout Attention * Grad [3] | 98.20%                 | 55.60%                    |
> | Ours | 99.20%                      | 88.80%                    |
> | Ours (w/ RS) | 99.20%                | **89.20%**                    |
>
> Our method outperforms the best saliency map based method (raw attention) by 7.8% on attack image accuracy.
>
> _Reducing spurious correlations_
>
> We conducted the removing spurious correlation experiment (Table 4) too with saliency maps. For raw attention and rollout attention, we mask out parts of the image with map > threshold with 0. For gradient based map, we take gradient respect to similarity to “this can be described as hair” and mask out parts of the image with map < threshold with 0. We report performance under the best threshold for each saliency map.
>
> | Method | Weighted Average ↑ | Male Gray Hair↑ | Male Non-Gray Hair↑ | Female Gray Hair↑ | Female Non-Gray Hair↑ |
> | --- | --- | --- | --- | --- | --- |
> | Baseline | 58.22% | 99.67% | 15.85% | 19.68% | 99.67% |
> | Raw Attention | 70.00% | 90.71% | 42.00% | 59.50% | 88.92% |
> | Raw Attention * Grad [1] | 66.17% | 99.10% | 19.01% | 49.52% | 98.86% |
> | Rollout Attention [2] | 69.95% | 91.61% | 38.27% | 59.98% | 91.12% |
> | Rollout Attention * Grad [3]| 62.48% | 99.43% | 25.20% | 28.53% | 98.78% |
> | Our Intervention | 81.66% | 97.23% | 74.01% | 58.00% | 98.29% |
> | Our Intervention (RS) | 83.91% | 97.23% | **74.80%** | **66.56%** | 97.80% |
>
> Our method outperforms best saliency map based method (raw attention) by 24.56% on worst class accuracy.
>
> [1] Selvaraju, R. R., Cogswell, M., Das, A., Vedantam, R., Parikh, D., & Batra, D. (2016). Grad-CAM: Visual Explanations from Deep Networks via Gradient-based Localization. arXiv preprint arXiv:1610.02391.
>
> [2] Samira Abnar and Willem Zuidema. Quantifying attention flow in transformers. In Proceedings of the 58th Annual Meeting of the Association for Computational Linguistics, pp. 4190–4197, Online, July 2020. Association for Computational Linguistics. doi: 10.18653/v1/2020.acl-main.385. URL https://aclanthology.org/2020.acl-main.385.
>
> [3] Hila Chefer, Shir Gur, and Lior Wolf. Transformer interpretability beyond attention visualization. In Proceedings of the IEEE/CVF Conference on Computer Vision and Pattern Recognition (CVPR), pp. 782–791, June 2021.
>
> **Advantage over interpreting token via gradient**
>
> Thank you for suggesting a comparison between our method and gradient-based method. If we understand the method you proposed correctly, given a token and a final layer CLS embedding, your proposed method calculates the gradient between the token and the cosine similarity between final layer CLS embedding and each text description. Then the text description for the token would be the description with the largest gradient. Please correct us if our understanding is inaccurate.
>
> Intuitively, your proposed method assigns X as token description if with a small change in token, the final layer CLS embedding ’s similarity to X will change a lot. One advantage of our method over this method is that our method finds a direct description for the latent token. In contrast, your proposed method might fail to find a direct description of the token. For example, a token that encodes “tire” might influences CLS’s similarity to “car” a lot, but your proposed method can only find “car” but not a direct description “tire” for the token.

---

> ### Author Response · Authors · 2023-11-22
> **Response continued**
>
> **Motivation for ''Intervening in Reasoning Procedure'' Experiment**
>
> While the saliency map overlap and causal intervention experiments in section 4.1 demonstrates that our interpretation is faithful, we hope to show that our interpretation also explains the causal influence between a token and final image representation. By replacing car tokens with airplane tokens and successfully changing final image representation from encoding highway to airport, we show that our interpretation not only explains what each token represents but also the causal relationship between a token’s and final image’s representation in ViT’s reasoning process.

---

### Author Response · Authors · 2023-11-22
**Updated paper draft to include comparison to other methods and additional usage example**

Thanks all reviewers for their valuable feedback. We have updated the paper draft and highlighted changes in blue text. Notably, we added comparison between editing performance based on our interpretation and that based on other interpretation methods. We also added an additional example of understanding adversarial attack with our interpretation framework in Appendix A.

---

### Meta-Review · Area_Chair_ogz1 · 2023-12-15

**Metareview:**

This work proposes a novel method for interpreting intermediate representations in Transformer-based vision-language models. Specifically, for a latent token at layer K the self-attention operations after that layer are removed and the last layer token representation is used to retrieve a text description which provides an interpretation of that latent token. The author hypothesize that by manipulating latent tokens the model behavior can be changed. This is an interesting and novel idea with an extensive experimental evaluation and an overall well written paper. To further improve the paper, I would suggest the authors include additional comparisons to related methods.

**Justification For Why Not Higher Score:**

Including additional baselines and a more thorough comparison to gradient-based methods would improve the paper

**Justification For Why Not Lower Score:**

Interesting and novel way of semantic attribution in vision foundation models

---

### Decision · Program_Chairs · 2024-01-16

Accept (poster)